# Evaluating the Impact of Heat Mitigation Strategies Using Added Urban Green Spaces during a Heatwave in a Medium-Sized City

**Nóra Skarbit, János Unger * and Tamás Gál** 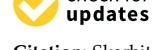

Department of Climatology and Landscape Ecology, University of Szeged, Egyetem u. 2, H-6722 Szeged, Hungary; skarbitn@geo.u-szeged.hu (N.S.); tgal@geo.u-szeged.hu (T.G.)

* Correspondence: unger@geo.u-szeged.hu

**Abstract:** Recognizing the growing trend of the urban population and the undeniable fact of global and regional climate change, it becomes increasingly important to explore how we can improve the livability of our cities not only in the distant future but also in the next few years. A critical aspect of this endeavor involves studying how we can effectively mitigate human heat load in urban areas. In our research, in the case of a medium-sized city (Szeged, Hungary), we examined the effect of surface modifications caused by vegetation on human thermal perception during the day and night of two heatwave days. To achieve this, we used the MUKLIMO_3 micro-scale climate model to simulate the thermal climate of Szeged, while the thermal load was assessed with the perceived temperature calculated by the Klima-Michel model. Our analysis also relied on the local climate zone (LCZ) system to describe the original land cover and the additional urban green spaces in the study area. We scrutinized the effects of added vegetation of different types and densities, as well as the presence of protective forests surrounding the city. Our findings revealed that the effect of the added vegetation can only be detected on the modified surfaces and in their immediate vicinity. Notably, dense urban greenery resulted in up to a 2–3 °C reduction in perceived temperature in certain areas during the daytime, highlighting the profound impact of targeted green space development. In addition, it is crucial to consider the airflow-blocking effect of woody vegetation, which can increase thermal load by 1–3 °C in the areas located in a downwind direction. Therefore, the changing regional climatic conditions (e.g., wind direction) and the development of the right type and location of urban green areas deserve special attention during modern urban planning processes.

**Keywords:** urban climate modeling; local climate zones; land cover modification; urban green spaces; human thermal perception

## 1. Introduction

Compared to the natural or agricultural environment around them, urbanized areas have different climatic characteristics, which means the development of a local climate called urban climate [1]. Depending on natural (geography, regional climate, etc.) and man-made (city size, structure, fabrics, and human activities, etc.) factors, this local climate can vary to a greater or lesser degree from city to city. However, it should also be added that in the case of a given city, there may be intra-urban differences in the extent of climate modification effects [2]. The climate changed by settlements leads to excessive thermal modification in the air of the urban canopy layer (urban heat island—UHI), the effect of which is especially pronounced at nighttime [3]. During warmer months, the UHI effect reduces outdoor thermal comfort and increases air-conditioning costs, adversely affecting the urban population [4]. Moreover, during multi-day extreme heat events (e.g., heatwaves), heat stress inevitably increases, which can cause an increase in morbidity and mortality due to heat stroke and hyperthermia [5–7].

In Europe, according to larger-scale and regional-scale climate change projections, summer temperatures will most likely increase in the 21st century, exacerbating heat stress

and thermal perception in urban areas, especially in cities, as shown by some Central European examples [8,9]. Urban green spaces can play a major role in reducing the heat load caused by built-up areas, thereby enhancing urban residents' well-being. Modeling such moderating effects of different types and locations of these and evaluating the results can provide fundamentally important information for urban planners. By providing evaporative cooling and shading, green areas help reduce high urban heat loads through these two processes [10]. On hot, dry summer days, human thermal sensation under tree canopies is more favorable than in open areas, as the leaves absorb and reflect a significant amount of incoming solar radiation [11–15]. It should also be noted that while higher vegetation can significantly reduce perceived temperatures, it generates greater aerodynamic drag, potentially impeding refreshing near-surface airflow [16]. This effect is influenced by the vegetation's density and vertical structure, with bushy vegetation less likely to hinder airflow compared to tall trees with expansive canopies, as evidenced by the model's detailed analysis of wind profile parameters. At night, however, the canopy can trap a significant part of the outgoing long-wave radiation, so the air below it cools less than, for example, above the grassy areas of the parks [12]. So, depending on their type and time of day, urban green spaces modify the urban heat load and thus the outdoor human thermal sensation level. Since outdoor human activity is primarily concentrated during the daytime hours, the daytime effect of green spaces in reducing heat load is significantly more important than in some cases their increasing effect at night. Therefore, it is generally agreed that the urban green environment helps to reduce the heat load as a whole [5]. Gál et al. [17] emphasizes that various types of urban green spaces have distinct influences on climate conditions, with their impacts on thermal perception and heat load mitigation assessed based on different climate indices. While these green spaces offer numerous benefits for urban well-being, our focus remains on their role in thermal comfort and heat load mitigation, without delving into other environmental aspects of them (e.g., that they are relatively quiet, relaxing, cleaner, and aesthetically refreshing areas).

The moderating effects of urban green areas can be quantified using various thermal indices. In the past, numerous simple thermal indices have been developed in order to describe the complex conditions of heat exchange between the human body and its thermal environment [18]. De Freitas and Grigorieva [19] have compiled a comprehensive catalog of human thermal climate indices proposed over the past 100 years. Their register assembles 162 thermal indices, and they are grouped according to eight classification classes (from single-parameter indices through those based on statistical models to indices based on the energy balance). The momentary outdoor human thermal sensation is always the result of the complex effect of the air temperature and humidity, the short- and long-wave radiation of the environment, and of wind conditions. The human condition must also be taken into account (activity level, clothing, posture, etc.). Therefore, only indices based on complete heat balance models are sufficient to evaluate the thermal environment in a thermo-physiologically significant way [18]. In other words, the analysis of intra-urban temperature differences is not suitable for detecting heat load differences between city districts in general, but especially not during heatwaves [2]. The perceived temperature (PT), as an energy balance stress index [19,20] offers a suitable approach for examining thermal comfort. Its utility is further underscored by its high ranking in a comprehensive evaluation of 165 human thermal climate indices [21].

Jendritzky et al. [18] described perceived temperature (PT) as a measure that equates the human sensation of cold or heat in a reference environment to the actual environmental conditions. This index is calculated using the Klima-Michel model, a comprehensive human body heat budget model. The model operates on the principles of the comfort equation established by Fanger [22], effectively translating various environmental factors into a unified perception of temperature. Regarding how widely PT is suitable for characterizing the bioclimatic environment (for example, for daily forecasts and the production of bioclimate maps at different scales), Jendritzky et al. [18] listed potential options and examples. As a later example, Kim et al. [23] studied the regional distribution of PT in South Korea using

long-term climate data based on synoptic observations. Staiger et al. [24] also confirmed the suitability of PT for many applications from micro to global, both in daily forecasts and in climatological studies.

There are several possibilities for modeling human thermal comfort parameters in urban areas; however, the most usual methods have limited spatial extent. As an example, in the case of ENVI-met [25] and SkyHelios [26], the model domain could cover several building blocks, and RayMan [27], as a widely used model, uses a 1D calculation approach. The novelty and potential of the application of MUKLIMO_3 for PT estimation of different urban green area layouts are the city-scale approach and the possibility for direct application for future climate scenarios using regional climate model outputs as forcing for the model, as [17] demonstrated in the case of basic heat-related climate indices.

Despite the potential applications and possibilities, research on PT patterns within cities or specific urban districts remains limited. Therefore, this study aims to address this gap by focusing on the influence of green areas in and around the city. The purpose of this study is to demonstrate the impact of different land cover modification scenarios on human perception under different wind directions. These scenarios are the following: more (a) dense trees, (b) scattered trees, (c) low plants, (d) mix of them in the city, and (e) protective forest around the city. To measure the human perception, perceived temperature (see Section 2.3) is used with particular regard to the different built-up areas of the city during a heatwave period in 2015 (04–14 August).

## 2. Data and Methods

### 2.1. Land Use Classification by Local Climate Zone System

The local climate zones (LCZ) system, developed by Stewart and Oke [27], was employed to depict the surface characteristics of the study area. This methodology is widely utilized in urban climatology research as it enables an objective differentiation of various urban and rural surfaces. The LCZ classes describe the climate-relevant aspects of the urban environment with ranges of values. Each urban category has associated variables that describe aspects of urban morphology and material properties in terms of value ranges, which are implementable in urban climate models (UCMs) [28,29]. LCZs became a standard in urban climate modeling [30].

The system distinguishes 17 types (referred to as zones). Among these, 10 of them are defined by the characteristics of their built-up area (LCZ 1-10) and 7 of them by the primary attributes of their land coverage (LCZ A-G). The naming of these zones is derived from specific surface parameters, including building height and density for built-up zones, and the nature and density of surface cover in the case of land cover zones (see the caption for Figure 1).

Different methodologies exist for identifying and mapping local climate zones within a given area [31]. Our study applied and adopted the approach developed by Bechtel et al. [32], which is particularly advantageous in scenarios with limited surface data. This methodology leverages freely available software, starting with the preliminary designation of zones using Google Earth, followed by the classification of these zones through the analysis of Landsat satellite images with SAGA-GIS software.

The obtained LCZ map of a city shows the character of the urban surface and, by association, the distribution of typical parameter values. Several studies provided evidence for the use of LCZ maps as a heat stress indicator [30,33,34]. Nevertheless, the final LCZ map should be treated as a general rather than a precise description of the layout and character of a city and its surrounding environment [28].

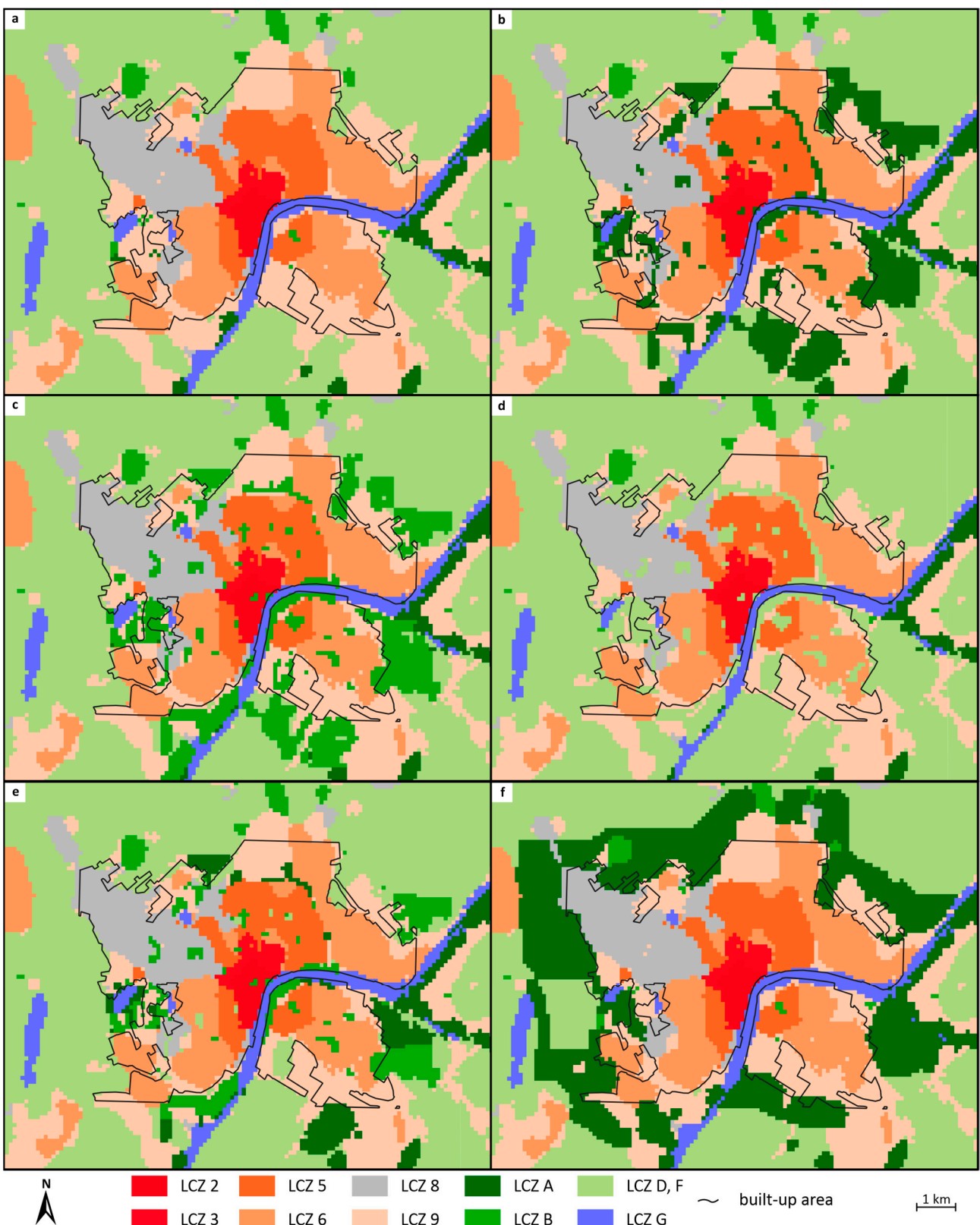

**Figure 1.** The original land cover characterized by LCZ scheme (**a**), and the modified land cover scenarios in Szeged, Hungary: dense trees (**b**), scattered trees (**c**), low plants (**d**), mixed vegetation (**e**), and protective forest (**f**) in the study area (the built-up urban area is delimited by the black line). Names of LCZs used in this study [27]: 2—compact midrise, 3—compact low-rise, 5—open midrise, 6—open low-rise, 8—large low-rise, 9—sparsely built, A—dense trees, B—scattered trees, D—low plants, F—bare soil or sand, G—water.

### 2.2. Study Area and the Investigated Land Cover Scenarios

Szeged is located in Central Europe, in a flat terrain of southeastern Hungary (46° N, 20° E) at an elevation of about 80 m above sea level. The region belongs to the Köppen's climatic type of Cfa (temperate climate with no dry season and warm summer) [35,36]. The city has 162,000 inhabitants and the urbanized area encompasses approx. 40 km². It encompasses a densely built center with mid-level height structures, openly arranged apartment blocks, extensive areas of single-family homes, as well as shopping malls and warehouses. River Tisza crosses the city from the northeast and divides it into two parts (Figure 1a). The neighboring rural area is mostly arable lands (e.g., maze, wheat), but in some places they are interrupted by scattered trees and groves [37].

The N–S and E–W extent of the study area is 10 km and 12.6 km, respectively (Figure 1). Within this area the built-up types in the city center are mainly compact zones (LCZs 2 and 3), which are surrounded by LCZ 5 (Figure 1a). The outlying parts of the city are covered by LCZs 6 and 9, while most of the northwestern part of the city is classified as LCZ 8.

The original land cover of the city, as a basic situation, was modified by 5 areal scenarios, through which various extra vegetation was added to the city and its surroundings. According to the applied surface classification (LCZ scheme), they are LCZs A, B, and D. In these classifications, there are no other details about the stands and about the types of vegetation. The thermal effects of different types and densities of vegetation (LCZs A, B, and D, Figure 1b–d) and, in one scenario, a combination of these on the delimited urban area were investigated (Figure 1e). In addition, a case in which the city is surrounded by a protective forest zone (LCZ A) can be particularly interesting (Figure 1f).

### 2.3. Thermal Sensation Modeling

In order to model the climatic conditions of Szeged, the MUKLIMO_3 model was applied, which is a non-hydrostatic micro-scale model and solves the Reynolds-averaged Navier–Stokes equations. The thermodynamically enhanced version of the model incorporates prognostic equations for atmospheric temperature and humidity, parametrizes the unresolved buildings, short-wave and long-wave radiation, and integrates balanced heat and moisture budgets in the soil and a vegetation model. The cloud processes, precipitation, horizontal runoff, and anthropogenic heat are not included in the model; however, it is capable of simulating the daily temperature cycle, relative humidity, and wind patterns at a high-resolution spatial scale [38,39]. In this study, the MUKLIMO_3 model was run on a horizontally equidistant grid with a resolution of 100 m, and the vertical resolution varies between 10 and 50 m, which is denser near to the surface. The initial meteorological data were provided by the ALARO model, and the station of the Hungarian Meteorological Service in Szeged was considered the reference station [40]. The modeled period was between 4 August and 14 August 2015, during which Central Europe experienced an extreme heatwave, with the maximum temperature exceeding 30 °C every day.

One of the post-processing tools of MUKLIMO_3 model was used to simulate the thermal sensation, which calculates the perceived temperature on the basis of the Klima-Michel model [23].

Staiger et al. [41] and Jendritzky et al. [18] elaborated on the concept of perceived temperature (PT), defined in terms of the air temperature (°C) of a given reference environment that would elicit the same sensation of heat or cold as the actual environmental conditions. In this reference setting, factors like wind velocity are minimized to a gentle draft, and the mean radiant temperature is the same as the air temperature, similar to conditions found in an extensive forest. Furthermore, the water vapor pressure is consistent with the actual environment unless condensation reduces it.

The calculation of perceived heat and cold is based on Fanger's comfort equation [22], which utilizes a comprehensive heat budget model for the human body. The assessment is made for a standard male figure named 'Klima Michel', characterized as 35 years old, 1.75 m tall, and weighing 75 kg. His assumed physical activity corresponds to walking at a speed

of 4 kmh$^{-1}$ on flat terrain, which means a work performance of 172.5 W. This assessment is specifically designed for outdoor conditions, allowing 'Klima Michel' the choice between summer and winter clothing in order to achieve the greatest possible thermal comfort.

The thermo-physiological meaning of PT values (°C) is presented in Table 1, where the level of thermal perception and thermo-physiological stress categories can be seen based on the range of the values. In its definition, the developers of PT did not mention the geographical limits of its applicability at all (nor in the case of the model used to calculate it). Thus, outside of Europe, PT was also used in, for example, South Korea [23]. Both countries (Germany and Hungary) are in Central Europe, geographically close to each other, so it is conceivable that there would be a shift of one or two degrees in the PT grades adapted to Hungarian conditions compared to the original ones. In our study, however, we did not examine the patterns of the absolute PT grades but the grade differences created by the added vegetation (that is, in which urban areas there is a shift in the heat perception categories due to cooling or warming). We can therefore use the categories of Table 1 to detect differences caused by different land cover scenarios.

**Table 1.** Thermo-physiological meaning of perceived temperature (PT) [24,41,42].

| PT Ranges (°C) | Thermal Perception Categories | Thermo-Physiological Stress Categories |
|---|---|---|
| $PT \geq 38$ | Very hot | Extreme heat stress |
| $32 \leq PT < 38$ | Hot | Great heat stress |
| $26 \leq PT < 32$ | Warm | Moderate heat stress |
| $20 \leq PT < 26$ | Slightly warm | Slight heat stress |
| $0 < PT < 20$ | Comfortable | Comfort possible |
| $-13 < PT \leq 0$ | Slightly cool | Slight cold stress |
| $-26 < PT \leq -13$ | Cool | Moderate cold stress |
| $-39 < PT \leq -26$ | Cold | Great cold stress |
| $PT \leq -39$ | Very cold | Extreme cold stress |

In order to comprehensively depict the impact of the mentioned land cover modification during the most intense heatwave conditions, we consider two days with different prevailing wind directions, specifically, northeast (NE) and south (S) and two distinct times: 14 LST (local summer time) and 1 LST. This approach provides a thorough understanding of how land cover changes affect daytime and nighttime thermal conditions under varying wind directions. The resulting figures (Figures 2–5) illustrate the spatial distribution of PT values in the case of the original land cover and the difference in PT maps resulting from changes in land cover. In these figures, we have marked the border of the PT categories that appear in the original situation and belong to the given time, as well as the areas of category change that occur as a result of different scenarios. Basically, we focus on the thermal changes in the built-up areas primarily used by the inhabitants (this urban area is delimited by a black line in Figure 1 and by a gray line in Figures 2–5).

Regarding the accuracy of the perceived temperature (PT) value simulations, no evaluations currently exist. However, Bokwa et al. [40] have demonstrated that MUKLIMO_3 is capable of precise temperature simulation in the same study area and under the same atmospheric forcing conditions.

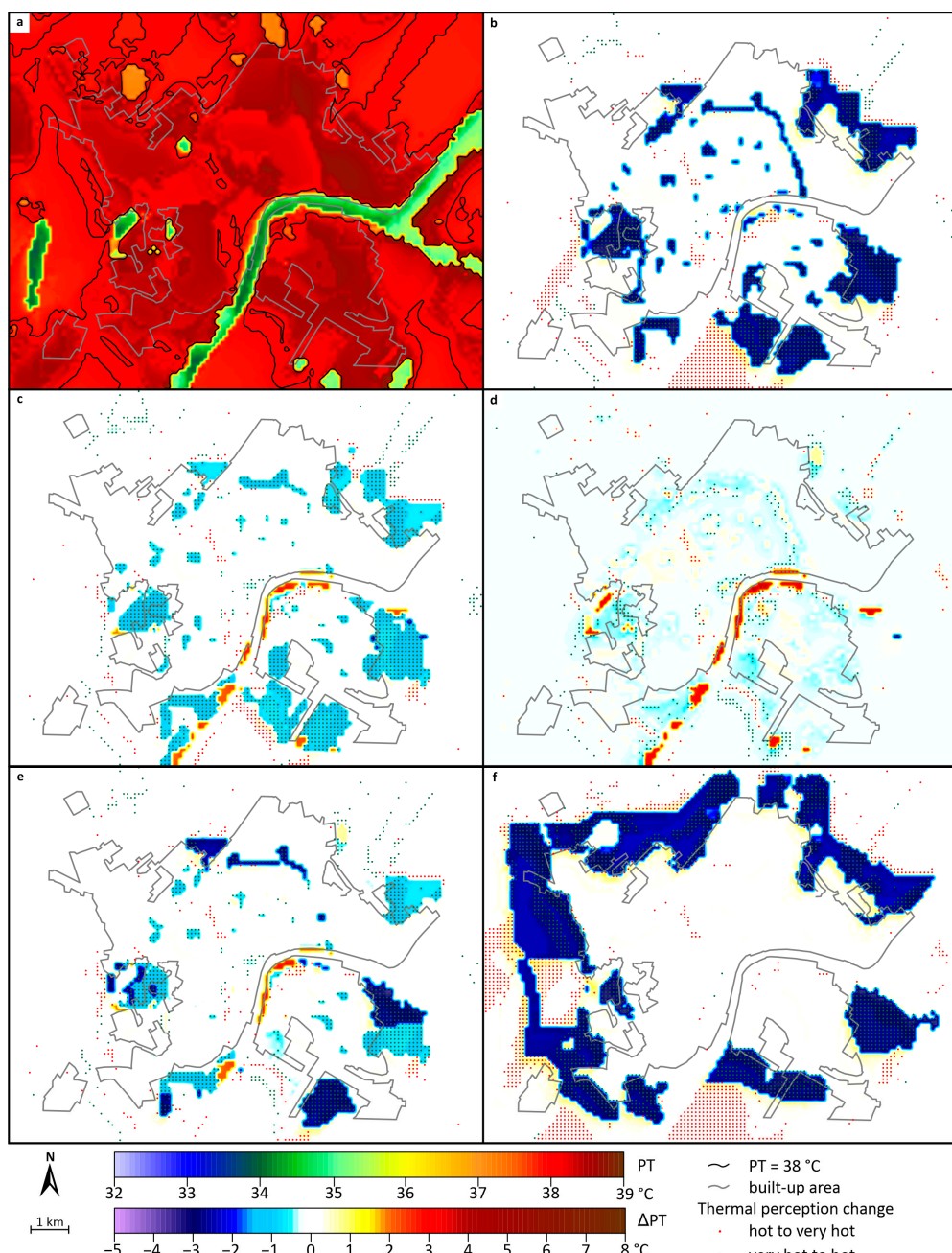

**Figure 2.** Pattern of perceived temperature (PT) of the original land cover (**a**) and PT differences (ΔPT) caused by land cover change ((**b**): LCZ A, (**c**): LCZ B, (**d**): LCZ D, (**e**): mixed, (**f**): protective forest) on 5 August 2015 at 14 LST (wind direction: NE) (black line marks the border between hot' and 'very hot' categories).

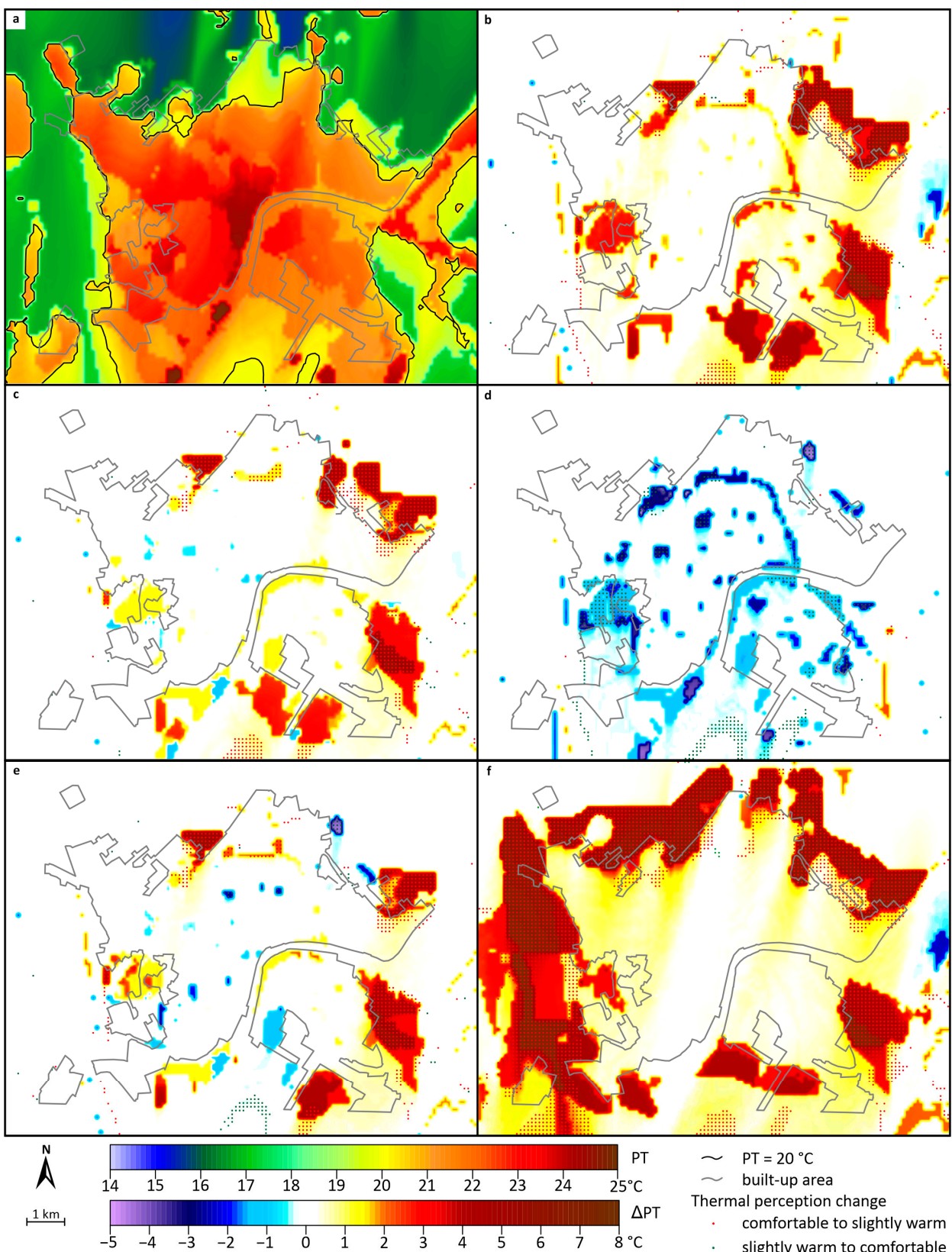

**Figure 3.** Pattern of perceived temperature (PT) of the original land cover (**a**) and PT differences (ΔPT) caused by land cover change ((**b**): LCZ A, (**c**): LCZ B, (**d**): LCZ D, (**e**): mixed, (**f**): protective forest) on 6 August 2015 at 1 LST (wind direction: NE) (black line marks the border between 'comfortable' and 'slightly warm' categories).

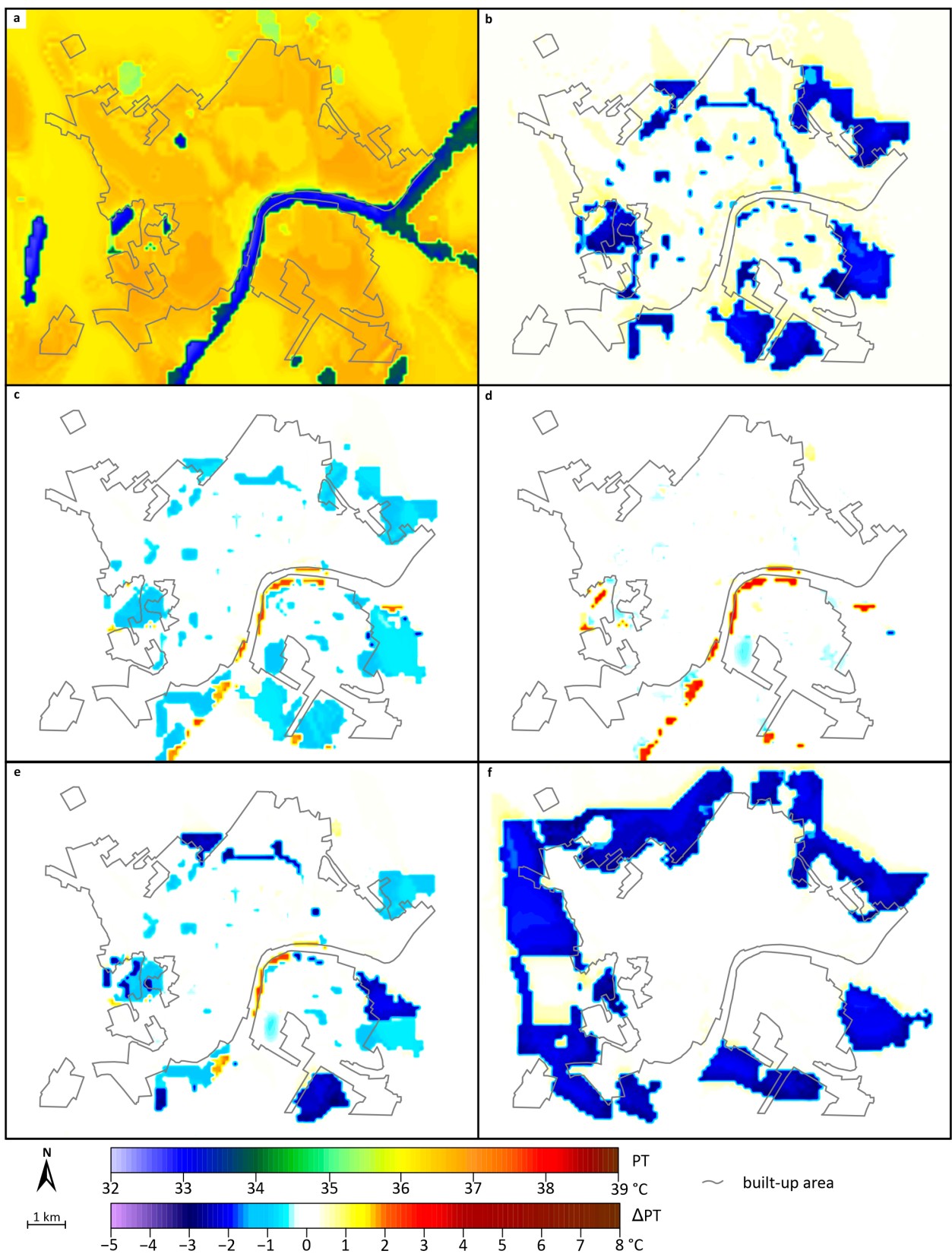

**Figure 4.** Pattern of perceived temperature (PT) of the original land cover (**a**) and PT differences (ΔPT) caused by land cover change ((**b**): LCZ A, (**c**): LCZ B, (**d**): LCZ D, (**e**): mixed, (**f**): protective forest) on 8 August 2015 at 14 LST (wind direction: S).

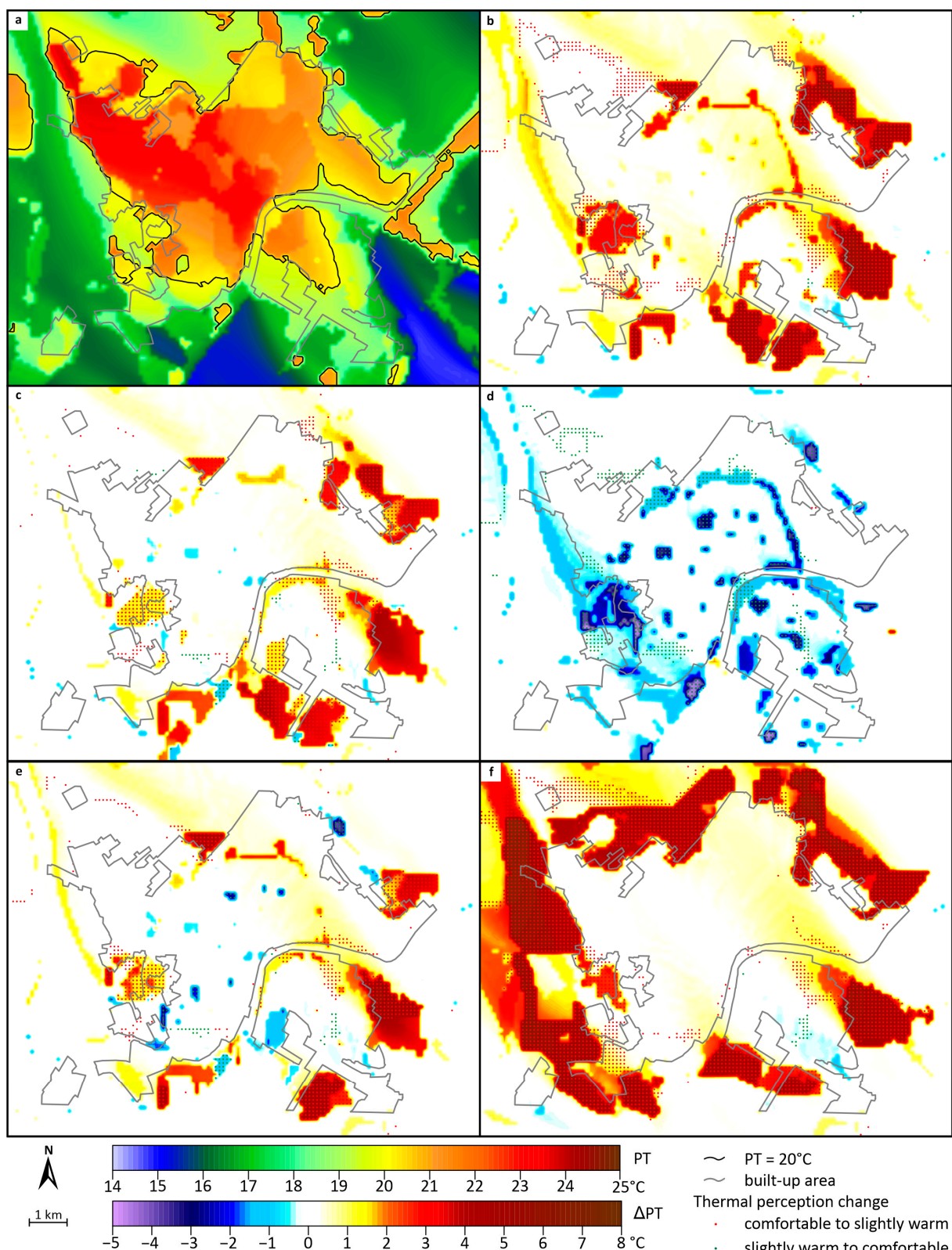

**Figure 5.** Pattern of perceived temperature (PT) of the original land cover (**a**) and PT differences (ΔPT) caused by land cover change ((**b**): LCZ A, (**c**): LCZ B, (**d**): LCZ D, (**e**): mixed, (**f**): protective forest) on 9 August 2015 at 1 LST. (wind direction: S) (black line marks the border between 'comfortable' and 'slightly warm' categories).

### 3. PT Pattern of Original Land Cover and Difference in PT Patterns under Different Land Cover Scenarios

*3.1. PT Patterns under NE Wind in Daytime (05.08.2015 at 14 LST)*

In the case of the original land cover, the daytime perceived temperature across the entire study area was notably high (Figure 2a). In rural regions, PT ranged between 37 and 38 °C ('hot' thermal perception and 'great heat stress'), but the largest part of the built-up area exhibited a bit higher values, typically varying between 38 and 39 °C (already 'very hot' and 'extreme heat stress'). Exceptions are a small LCZ 6 area and around a small lake to the northwest of the center, as well as the areas along the river, where only 'hot' perception is experienced. The displacing effect of the NE wind is barely noticeable, mostly only in the SW part of the city, where the heat load is therefore slightly greater than in the central area.

The main effect of the addition of dense tree (LCZ A) cover is most pronounced primarily in the modified areas and their immediate vicinity (Figure 2b), reducing the heat load by 2–3 °C. This reduction only relates to the 'hot' thermal perception in the built-up parts, where the newly treed areas are, although only in patches. Due to the wind-blocking effect of the trees, a slight rise in PT (<1 °C) can be observed within the city, as well as only in smaller spots (already 'very hot' category).

The pattern obtained as a result of the land cover modification by scattered trees (LCZ B) is very similar to the dense tree case (Figure 2c), although the PT change is more moderate. The category changes mentioned above still occur in thermal perception, but also only in spots, and the cooling effect ('very hot' to 'hot') is dominant in these spots.

The effect of adding low vegetation (LCZ D) is hardly noticeable in the urban area— the change is typically less than half a degree (Figure 2d)—and also only in small areas, which results in a change of one category in both directions, but mainly the cooling effect is dominant. Areas close to water (along the eastern bank of the Tisza river) are exceptions, where the additional low plants lead to more than a 2 °C increase in thermal load.

In the scenario of mixed land cover modification, the introduction of vegetated areas results in an intermediate effect compared to the previously mentioned cases, inducing both warming and cooling phenomena (Figure 2e). The result is that the 'very hot' category also appears in some smaller northern and northeastern patches, while on the eastern and western edge of the city, there is a slight reduction in the heat load ('very hot' to 'hot').

The protective forest practically has no noticeable effect within the city (Figure 2f). An exception is a very small LCZ 6 area to the northwest of the center, where the originally 'hot' category changes to 'very hot'. As a result, the 'very hot' thermal sensation dominates the entire urban area in the case of this scenario.

*3.2. PT Patterns under NE Wind at Night (06.08.2015 at 1 LST)*

In the case of the original land cover, the nocturnal PT values range from 18 to 23 °C, but values above 20 °C are dominant. Therefore, almost the whole urban area has 'slightly warm' thermal perception except for the 'comfortable' LCZ 6 and LCZ 9 areas in the furthest north (Figure 3a). PT values are higher in the central and SE parts of the city than in the NE parts, presumably due to the prevailing NE wind. These findings indicate that there is a persistent 'slight heat stress' at night in the city during heatwaves (Table 1).

Generally, the introduction of dense tree cover (LCZ A) tends to raise the PT values within the city by 1–3 degrees. Due to the NE wind, the affected areas clearly can be recognized, extending southwest from the new land cover (Figure 3b). All this results in a category step in the northern and northeastern parts of the urban area, where originally 'comfortable' conditions prevailed. So, for this surface scenario, a 'slightly warm' thermal perception is experienced throughout the urban area during the night.

In the case of change by scattered trees (LCZ B), changes similar to those achieved by dense tree cover are realized in the same areas, although to a more modest extent (Figure 3c). The PT values increases by 1–2 degrees within the city, and the effect of the NE wind can also be recognized. A reduction in PT ranging from 0.5 to 1 °C can be intermittently

observed in various locations. The result is that the 'slightly warm' category also appears in some smaller northern and northeastern areas.

The added low plant areas (LCZ D) result in reduced thermal load, but their effect is only apparent in the modified areas (Figure 3d). Here, this decline typically falls within the range of 1–2 °C, but in some places it can exceed 3 or even 4 °C. Thus, the affected southeastern places and the area to the west are moved from the 'slightly warm' to the 'comfortable' category.

In the case of the mixed land cover scenario, the added vegetated areas lead to an intermediate situation regarding the effects of the previous three scenarios, mainly causing some warming (Figure 3e). The result is that the 'slightly warm' category also appears in some smaller northern and northeastern patches.

The presence of protective forest (LCZ A) surrounding the city leads to a substantial and extensive increase in thermal load (Figure 3f). The most remarkable alterations occur in the northern and northeastern urban parts, where the PT changes by more than 3 °C, thereby turning the originally 'comfortable' into a 'slightly warm' thermal perception in these areas. However, it should be noted that these large thermal changes are primarily limited to areas near the altered surface cover. Within the city, the effect of the forest only means an increase in PT between 0 and 2 degrees, so it does not change the original thermal categories here.

### 3.3. PT Patterns under S Wind in Daytime (08.08.2015 at 14 LST)

Two days later, the daytime heat load was a bit lower compared to the previously presented day, and there were only small variations in thermal sensation within the study area (Figure 4a). Except for a few small spots (areas near the water surface, 33–35.5 °C), in most parts of urban areas, PT values ranged between 36 and 37 °C, indicating 'great heat stress' ('hot' thermal perception, Table 1). Within the built-up areas, PT mostly exceeded 36.5 °C, with the highest values occurring in LCZs 6, 8, and 9. Conversely, in LCZ 5 and the downtown areas of LCZs 2 and 3, the values were slightly lower.

The increase in dense tree cover (LCZ A) results in a 2 °C decrease in PT, but this change is only observed in the modified regions (Figure 4b). These alterations in land cover do not influence thermal perception in the surroundings of these areas, even in the immediate vicinity. It is also interesting that a significant part of the urban area shows an increase of about 0.5 degrees. As a result, the heat load of this period decreases only in the changed areas, but it is still burdensome and remains in the 'hot' category (Table 1).

Scattered trees (LCZ B) and, similarly, low plant (LCZ D) cover in the modified areas only result in a modest decrease of 0.5–1 °C, while on the banks of the River Tisza, heat perception increases by about 1 degree (Figure 4c,d). All of these do not cause any category change in the urban area.

Considering the mixed land cover scenario, in fact, in accordance with the results of the previous three scenarios, there is no change in the daytime heat perception either (Figure 4e).

The protective forest that surrounds the city, presumably due to the blockage of air flow, has no noticeable PT-changing effect within the city, meaning that the entire urbanized area remains in the 'hot' category (Figure 4f).

### 3.4. PT Patterns under S Wind at Night (09.08.2015 at 1 LST)

On this night at 1 LST, most areas of the city experienced PT values above 20 °C ('slightly warm' thermal perception, Figure 5a). Then, there are smaller or larger PT differences between the different built-up areas: LCZs 5, 6, and 9 have lower PTs (between 20 and 22 °C), while LCZs 2, 3, and 8 show higher PTs above 22 °C. The influence of the S wind can be recognized in the higher values in the northwestern parts of the city (mainly LCZ 8).

The added dense tree cover (LCZ A) leads to an increase in thermal load by 1–3 degrees in some areas within the city, mainly in the northeastern (LCZ 5) and in the southern (LCZ 6)

parts, where thermal perception changes from 'comfortable' to 'slightly warm' category (Figure 5b).

The scattered tree cover (LCZ B) scenario results in a similar, albeit smaller, thermal load increase in roughly the same areas as the previous one, where even this smaller increase is sufficient for the category changes mentioned above to occur (Figure 5c).

A contrasting outcome can be observed when low plants (LCZ D) are added, leading to a decrease in thermal load, but its effect occurs primarily in the modified areas, and to a small extent, downwind from them (Figure 5d). The decrease is usually between 1–2 °C, but in some places it exceeds 3 °C, as a result of which in the affected southern and southeastern locations (LCZ 8) as well as in some northern areas (LCZ 9), the thermal load on residents is relieved ('slightly warm' to 'comfortable').

Considering the mixed land cover scenario, similar to the night situation discussed above, the areas with added vegetation lead to an intermediate situation in terms of the effects of the previous three scenarios, mainly causing some warming (1–3 °C) in the changed areas but also causing cooling (1–2.5 °C) in some places, depending on the type of vegetation. The effect of the tree-covered surfaces is also evident in the downwind direction, while the impact of the low plants land cover is more localized. As a result of all this, originally 'comfortable' but now 'slightly warm' areas appear in the west and south (LCZ 9) as well as in the east (LCZ 6), while also in the south some smaller, originally 'slightly warm' areas (LCZ 6) are reclassified as 'comfortable'.

Similar to the situation the night before, the protective forest primarily increases the thermal load in the northeastern part of the urban area (1–2 °C). In this case, it can be attributed to the S wind, but it causes a thermal perception category change ('comfortable' to 'slightly warm') only in the southern areas (LCZs 6 and 9) (Figure 5f).

## 4. Discussion and Conclusions

This study presents a thorough analysis of how variations in green land cover influence perceived temperature (PT) in Szeged, Hungary, throughout the daytime and nighttime of a heatwave period (4–14 August 2015). The results highlight the vital importance of urban green spaces in reducing the urban heat island (UHI) effect and improving thermal comfort amid heatwave conditions in medium-sized cities, such as Szeged, Hungary. With urban expansion and the rise in extreme heat events due to climate change, the strategic integration of green spaces into urban planning and policy becomes increasingly crucial for fostering sustainable urban development and enhancing public health.

This investigation pioneers the integration of city-wide human comfort simulations, utilizing the applied model on the basis of the LCZ system to elucidate the complex interplay between urban vegetation and thermal comfort. The methodology adopted herein not only provides granular insights into the cooling effects of urban green infrastructure but also provides the potential for incorporating outputs from regional climate models. This fusion could provide basic information about future urban thermal comfort assessments and evidence-based urban planning and climate adaptation strategies. The capability of the model to assimilate outputs from regional climate models positions this study as an exemplary foundation for future assessments of human comfort. By applying this methodology, there is the potential to significantly enhance our ability to predict and mitigate urban heat island effects, thereby bolstering urban resilience and sustainability against the backdrop of climate change.

This study underscores insights into urban microclimate management, particularly in the context of climate change adaptation and urban development challenges. The findings reveal how land cover modifications, such as the introduction of tree cover and low vegetation, can affect human thermal perception (measured by perceived temperature) in urban areas, both during daytime and nighttime in a heatwave period. These outcomes highlight the complex interplay between urban planning, green infrastructure, and microclimatic conditions, which are essential components in addressing urban development and climate change challenges.

According to the results, the vegetation-based changes in land cover, such as increasing areas of tree cover, low vegetation, and protective forest, can have a noticeable impact on PT in both urban and rural areas. These surface modifications can lead to remarkable differences in thermal perception between the different neighborhoods of the study area, knowledge of which can be particularly important during heatwaves.

During the day, the original land cover showed particularly high PT values in the study area, with minimal variations between rural and built-up areas (practically above 36–37 °C, and the load even exceeds 38 °C on the first day, i.e., 'very hot' in large areas. The introduction of dense trees can reduce PT by up to 2 °C in urban centers and by up to 3 °C in less populated areas. However, dense trees can also obstruct airflow, leading to PT increases on the downwind side. The addition of low vegetation has a less discernible impact in built-up areas, but it leads to an increase in thermal perception of 2–3 °C near the outskirts and bodies of water. Various effects in the mixed land cover demonstrate that the nature and location of the modification significantly influence the obtained PT values. The protective forest, presumably due to the blockage of air flow, practically has no noticeable effect within the urban area, meaning that the entire area remains in the 'hot' or 'very hot' category.

Originally, nighttime PT values indicated a slight heat load in the city during the heatwave, with notable variations between urban parts. The addition of dense tree cover increases the PT values within the city by 1–3 °C, and on the first night, due to the NE wind, the affected areas can be clearly identified, extending southwest from the new land cover. Scattered tree cover leads to changes similar to those achieved by dense tree cover but with a moderate increase in heat load. The low plant scenario results in a reduction in the thermal load (1–2 °C), mostly in the modified regions, but in some places it can exceed 3 or even 4 °C. The mixed land cover results in an intermediate situation in terms of the effects of the previous three scenarios, mainly causing some warming (1–3 °C), but also cooling (1–2.5 °C) depending on the type of vegetation. The presence of the protective forest surrounding the city leads to an increase in thermal load (1–3 °C), but this change is primarily limited to areas near the altered surface cover.

This study demonstrates that dense tree cover and protective forest can significantly reduce PT during the day, thus offering a strategic way to mitigate daytime heat load, although they have a territorially limited effect. On the other hand, their effect on the thermal conditions at night is more nuanced. The results suggest that while trees, especially denser plantings, are beneficial during the day, their effects can be complex at night, as their canopy can potentially trap some of the heat radiated from the surface.

The results of this study emphasize the intricate effects of land cover changes on microclimate dynamics, which are shaped by factors such as wind direction. Specifically, while dense tree coverage can decrease perceived temperatures (PTs) during the daytime, it might also hinder airflow, leading to increased PTs in downwind areas. This underscores the necessity for spatial planning that incorporates considerations of natural airflows and urban structure to maximize the cooling benefits of green spaces. Furthermore, the findings advocate for an evidence-based strategy in urban green space development, acknowledging the complex interplay between vegetation, urban morphology, and climatic conditions. Urban planning policies should leverage insights from microclimate modeling studies, such as this one, to inform the design of green infrastructure scenarios that optimize thermal comfort. Such a methodological approach can direct resource allocation towards the most effective interventions, ensuring that green space development is both efficient and contributes to the overall thermal well-being in urban areas.

Nighttime cooling is essential for human health and well-being, providing necessary respite from daytime heat exposure, with the persistence of nocturnal thermal load during heatwaves posing significant public health risks. City planners face the challenge of leveraging the daytime cooling benefits of trees while mitigating their potential to trap heat at night, necessitating strategic tree placement and the optimization of urban green spaces to enhance their cooling effects. Beyond environmental and health considerations, the socio-

economic aspects of urban green space distribution also demand attention. Equitable access to these spaces is critical for alleviating heat stress among all urban dwellers, especially vulnerable populations such as the elderly, children, and those in economically disadvantaged areas. Consequently, policies aimed at increasing urban greenery should address social equity, ensuring widespread accessibility to the advantages of green spaces. This may include targeted efforts in underserved communities, engaging residents in the design and upkeep of green spaces, and promoting their multifunctional use for recreation, social interaction, and cultural enrichment, thereby harmonizing environmental sustainability with social well-being.

The findings of this study are pivotal for the trajectory of future urban development, highlighting the imperative for urban planners and developers to incorporate microclimatic considerations into the expansion and densification of cities. This necessitates a deep understanding of how different land covers influence local thermal loads and the strategic use of these covers to counteract the adverse effects of climate change, particularly in densely populated areas. Elevating the proportion of green infrastructure is crucial for enhancing urban environments and sustainability, making land cover modification a key strategy in urban planning to alleviate heat load and improve thermal perception amidst the escalating regional warming of the present and future.

This study represents a significant leap in comprehending how urban green spaces can be effectively utilized to mitigate heat stress during extreme heat events in medium-sized cities. Through the application of this model combined with the PT index, it provides detailed insights into how different types of vegetation impact urban thermal comfort. This research moves beyond a focus on large urban areas, spotlighting the specific challenges and opportunities for medium-sized cities like Szeged, Hungary, in adapting to climate change. This innovative approach not only deepens our understanding of the microclimatic benefits of urban greenery but also establishes a new standard for future urban climate modeling, emphasizing the importance of the LCZ system in urban heat management.

The implications of this research are extensive, bridging the gap between basic research and the fields of urban planning, public policy, and community health. By delivering concrete evidence of the cooling efficacy of green infrastructure, the study equips urban planners and policymakers with a crucial tool for enhancing city resilience against global warming. It offers actionable recommendations on the spatial distribution and varieties of vegetation that achieve the most effective cooling, guiding the creation of more sustainable and habitable urban spaces. As cities globally confront the growing challenges of climate change and urban heat islands, the insights provided here could fuel worldwide initiatives to integrate green infrastructure into urban climate adaptation strategies, thereby bolstering the health, well-being, and sustainability of urban communities.

**Author Contributions:** All authors contributed to the study conception and design. Material preparation, data collection, and analysis were performed by N.S. and T.G. The first draft of the manuscript was written by N.S. Review and editing of the manuscript was conducted by J.U. and T.G. All authors commented on previous versions of the manuscript. All authors have read and agreed to the published version of the manuscript.

**Funding:** The study was funded by the Hungarian Scientific Research Fund (OTKA K-137801 and PD-143378). Project no. TKP2021-NVA-09 has been implemented with the support provided by the Ministry of Culture and Innovation of Hungary from the National Research, Development and Innovation Fund, financed under the TKP2021-NVA funding scheme.

**Institutional Review Board Statement:** Not applicable.

**Informed Consent Statement:** Not applicable.

**Data Availability Statement:** The data presented in this study are available on request from the corresponding author.

**Acknowledgments:** We acknowledge the German Meteorological Service (DWD) for providing access to the MUKLIMO_3 model.

**Conflicts of Interest:** The authors declare no conflicts of interests.

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
