# Peer review of "Evaluating the Impact of Heat Mitigation Strategies Using Added Urban Green Spaces during a Heatwave in a Medium-Sized City"

_sustainability, doi:10.3390/su16083296_

Round 1

Reviewer 1 Report

Comments and Suggestions for Authors

The authors evaluated the impact of heat mitigation strategies using added urban green spaces. The research methodologies are reasonable, and the findings are interesting. However, there are many work related to this topic, and the biggest flaw is the innovation. There are still some aspects that should be improved to make the paper publishable. I focus here only on some points, which are hopefully easy for the authors to take into account in the revision.

(1) Part Introduction - the biggest flaw is the innovation, and it should be highlighted. Importantly, hightlight the added value to to emphasize the scientific contribution. Furthermore,  why choose LCZ?

(2) Part Data and Methods - it is good to add data sources and more details related to the study area (figure).

(3) Check Table 1 - '+'?

(4) How to ensure the accuracy of model simulation?

(5) Discussion - it shoud be discussed in depth.

Comments on the Quality of English Language

Minor editing of English language required.

Author Response

Comments and Suggestions for Authors

The authors evaluated the impact of heat mitigation strategies using added urban green spaces. The research methodologies are reasonable, and the findings are interesting. However, there are many work related to this topic, and the biggest flaw is the innovation. There are still some aspects that should be improved to make the paper publishable. I focus here only on some points, which are hopefully easy for the authors to take into account in the revision.

(1) Part Introduction - the biggest flaw is the innovation, and it should be highlighted. Importantly, hightlight the added value to emphasize the scientific contribution. Furthermore, why choose LCZ?

Explanation is added to the end of the introduction in order to clearly articulate the novelty of this work (Lines 100-107).

Regarding the use of the LCZ system we added the next sentences and references to the text:

“The LCZ classes describe the climate relevant aspects of the urban environment with ranges of values. Each urban category has associated variables that describe aspects of urban morphology and material properties in terms of value ranges, which are implementable in urban climate models (UCMs) [28–29]. Therefore, the LCZs became a standard in urban climate modelling as they capture important urban morphological characteristics [30].” (Lines 126-130 and citations 28-30)

„The obtained LCZ map of a city shows the character of the urban surface and, by association, the distribution of typical parameter values. Several studies provided evidence for the use of LCZ maps as a heat stress indicator [e.g. 30, 33–34]. Nevertheless, the final LCZ map should be treated as a general rather than a precise description of the layout and character of a city and its surrounding environment [28]”. (Lines 143-147 and citations 28, 30, 33-34)

[28] Brousse, O.; Martilli, A.; Foley, M.; Mills, G.; Bechtel, B. WUDAPT, an efficient land use producing data tool for mesoscale models? Integration of urban LCZ in WRF over Madrid. Urban Climate 2016, 17, 116–134. 10.1016/j.uclim.2016.04.001

[29] Hammerberg, K.; Brousse, O.; Martilli, A.; Mahdavi, A. Implications of employing detailed urban canopy parameters for mesoscale climate modelling: a comparison between WUDAPT and GIS databases over Vienna, Austria. Int. J. Climatol. 2018, 38/S.1, e1241–e1257. 10.1002/joc.5447

[30] Verdonck, M-L.; Demuzere, M.; Hooyberghs, H.; Beck, C. Cyrys, J.; Schneider, A.; Dewulf, R.; Van Coillie, F. The potential of local climate zones maps as a heat stress assessment tool, supported by simulated air temperature data. Landsc. Urban Plan. 2018, 178, 183–197, 10.1016/j.landurbplan.2018.06.004

[33] Geletič, J.; Lehnert, M.; Savić, S.; Milošević, D. Modelled spatiotemporal variability of outdoor thermal comfort in local climate zones of the city of Brno, Czech Republic. Sci. Total Environ. 2018, 624, 385 – 395. 10.1016/j.scitotenv.2017.12.076

[34] Giannaros, C.; Agathangelidis, I.; Papavasileiou, G.; Galanaki, E.; Kotroni, V.; Lagouvardos, K.; Giannaros, T.M.; Cartalis, C.; Matzarakis, A. The extreme heat wave of July–August 2021 in the Athens urban area (Greece): Atmospheric and human-biometeorological analysis exploiting ultra-high resolution numerical modeling and the local climate zone framework. Sci. Total Environ. 2023, 857, 159300. 10.1016/j.scitotenv.2022.159300

(2) Part Data and Methods - it is good to add data sources and more details related to the study area (figure).

Figure 1. presents the study area, and basic climate, geographical information is presented including a self-citation for further information. For us it is not clear what kind of further information is necessary for the solid evaluation of the results of this study.

(3) Check Table 1 - '+'?

The marks of ‘+’ are deleted in Table 1.

(4) How to ensure the accuracy of model simulation?

            Explanation is added to the end of Chapter 2.3. (lines 241-243)

(5) Discussion - it should be discussed in depth.

            The first 2 and last 5 paragraphs of Chapter 4. Discussion and conclusions are highly elaborated in order to provide more deep discussion of the paper. (lines 394-413 and 457-512)

Comments on the Quality of English Language

Minor editing of English language required.

Thank you for your feedback on the English language quality of our manuscript. We've carefully reviewed and refined the language to the best of our abilities now and prior to submission. While we aim for the highest standards, we recognize there may be minor imperfections. We believe the scientific content is communicated effectively and clearly.

Reviewer 2 Report

Comments and Suggestions for Authors

Most of the work is very good. The paper shows how climate change is changing the quality of life of urban residents. The authors suggest that when planning green infrastructure, much attention should be paid to the effect of blocking wind flow by woody vegetation, which can increase the heat load in areas located in the direction of the wind.

The article only needs major improvements in the discussion section, as the authors unfortunately do not refer to the work of other authors in any way, nor do they compare their research with the work of other researchers. 

Author Response

Comments and Suggestions for Authors

Most of the work is very good. The paper shows how climate change is changing the quality of life of urban residents. The authors suggest that when planning green infrastructure, much attention should be paid to the effect of blocking wind flow by woody vegetation, which can increase the heat load in areas located in the direction of the wind.

The article only needs major improvements in the discussion section, as the authors unfortunately do not refer to the work of other authors in any way, nor do they compare their research with the work of other researchers.

The first 2 and last 5 paragraphs of Chapter 4. Discussion and conclusions are highly elaborated in order to provide more deep discussion of the paper. In addition, a new paragraph (second in Chapter 4) is added in order to express the pioneering role of this study.

Reviewer 3 Report

Comments and Suggestions for Authors

Abstract

Line 13

Is the inclusion of two heat-wave days sufficient in order to generate representative results?

Line 14

In the Abstract section, the nature of the MUKLIMO_3 model should be stated. It is not safe to assume that the readers knowledge this model.

Line 19 to 20

Readers are attracted to quantitative findings. Please supplement the quantitative magnitude of the effect of adding vegetation.

Introduction

Line 37

UHI effect is especially pronounced at nighttime. This fact should be mentioned here.

Line 51 to 54

The change in drag can be illustrated by citing the observed surface roughness values in different studies.

Line 78 to 86

The relevance of parameters should be evaluated. Please state the parameters whose significance is high and thus used in this study.

Data and Methods

Line 109 to 125

This section is actually well-written. This level of detail can be applied to other sections as well.

Line 141 to 146

The five chosen scenarios can be elaborated. For instance, in the fifth scenario, the urban forest stand characteristics can be described.

Line 186 to 188

In Table 1, it seems that the thermal perception categories were developed in a German context. It is understood that Hungaria and Germany are close. Yet, the categorisation of thermal perception is specific to the geographical location and the adaptation of thermal conditions. Justifications should be provided on the categorisation scheme.

Results

Line 207 to 209

Throughout this section, it can be seen that the temperature ranges of different local climate zones were compared. But the comparison is only meaningful if the significance of the difference is elaborated. Related statistics can be provided.

Line 320 to 351

Echoing to my comments above, the description about the forest characteristics is too simple here. More quantitative parameters regarding the spatial distribution and ecological composition of the forest stands are required.

Discussion and conclusion

Line 353 to 425

I have no critical comments on this section.

Comments on the Quality of English Language

The language can be polished in order to enhance the readability of the manuscript. A lot of abbreviations are used throughout the manuscript. The authors should check whether the full form of all of the abbreviations are provided.  

Author Response

Comments and Suggestions for Authors

- Abstract

Line 13

Is the inclusion of two heat-wave days sufficient in order to generate representative results?

The model was run for a two-week heatwave period, but the analysis of the entire period would be too long due to space limitations and comprehensibility. Therefore, we selected the daytime and nighttime cases of 2 typical extreme days, which still meant 4 large figures and the related analysis. In our opinion, this is sufficient to illustrate the thermal sensation effect of added vegetation during such an extreme period.

Line 14

In the Abstract section, the nature of the MUKLIMO_3 model should be stated. It is not safe to assume that the readers knowledge this model.

The micro-scale and climate characteristics have been added to the line 14, details of the model are explained later in the text.

Line 19 to 20

Readers are attracted to quantitative findings. Please supplement the quantitative magnitude of the effect of adding vegetation.

            A new sentence is added, and one is modified in order to provide quantitative findings in abstract (lines 20-22 and 24).

- Introduction

Line 37

UHI effect is especially pronounced at nighttime. This fact should be mentioned here.

Added to the text (lines 38-39).

Line 51 to 54

The change in drag can be illustrated by citing the observed surface roughness values in different studies.

A more detailed explanation and a reference is added to the mentioned part of the paper (citation 16).

Shaw, R.H; Pereira, A.R. Aerodynamic roughness of a plant canopy: a numerical experiment. Agricultural Meteorology, 1982, 26(1), 51-65. 10.1016/0002-1571(82)90057-7

Line 78 to 86

The relevance of parameters should be evaluated. Please state the parameters whose significance is high and thus used in this study.

The purpose of this study was not to investigate the significance of different parameters. Incidentally, this was already done exhaustively by Staiger et al. [24].

- Data and Methods

Line 109 to 125

This section is actually well-written. This level of detail can be applied to other sections as well (citation 16 and line 54-59)

            Thank you for the appreciation.

Line 141 to 146

The five chosen scenarios can be elaborated. For instance, in the fifth scenario, the urban forest stand characteristics can be described.

According to the applied surface classification (LCZ scheme), they are LCZs A, B and D. In these classification there are no other details about the stands and about the types of vegetation.

Added to the text (lines 166-168).

Line 186 to 188

In Table 1, it seems that the thermal perception categories were developed in a German context. It is understood that Hungaria and Germany are close. Yet, the categorisation of thermal perception is specific to the geographical location and the adaptation of thermal conditions. Justifications should be provided on the categorisation scheme.

In its definition, the developers of PT did not mention the geographical limits of its applicability at all (nor in the case of the model used to calculate it). Thus, outside of Europe, PT was also used in, for example, South Korea [22]. Indeed, both countries (Germany and Hungary) are in Central Europe, geographically close to each other. It is conceivable that there would be a shift of one or two degrees in the PT-grades adapted to Hungarian conditions compared to the original ones. In our study, however, we did not examine the patterns of the absolute PT-grades, but the grade-differences created by the added vegetation (that is, in which urban areas there is a shift in the heat perception categories due to cooling or warming). We can therefore use the categories of Table 1 to detect differences caused by different land cover scenarios.

Added to the text (lines 216-225).

- Results

Line 207 to 209

Throughout this section, it can be seen that the temperature ranges of different local climate zones were compared. But the comparison is only meaningful if the significance of the difference is elaborated. Related statistics can be provided.

Thank you for your insightful remark regarding the comparison of temperature ranges across different local climate zones and the importance of elaborating on the significance of these differences. Due to the nature of the data and the modeling approach used in this study, direct calculation of statistical significance for the differences observed between local climate zones was not feasible. The methodology primarily focused on micro-scale climate modeling and Perceived Temperature calculations, which, while providing valuable insights into thermal perception variations, does not lend itself to traditional statistical analysis of significance typically associated with observational data sets.

Line 320 to 351

Echoing to my comments above, the description about the forest characteristics is too simple here. More quantitative parameters regarding the spatial distribution and ecological composition of the forest stands are required.

 According to the applied surface classification (LCZ scheme), the added vegetation as land cover are LCZs A, B and D. In these classification there are no other details about the stands and about the types of vegetation).

Added to the text (lines 166-168).

- Discussion and conclusion

Line 353 to 425

I have no critical comments on this section.

Thank you for your remark.

- Comments on the Quality of English Language

The language can be polished in order to enhance the readability of the manuscript. A lot of abbreviations are used throughout the manuscript. The authors should check whether the full form of all of the abbreviations are provided. 

The full form of all of the missing abbreviations are provided (lines 135-136 and lines 176-178). Some abbreviations have been moved as they appeared too early in the text before revision (lines 116-117, as well as lines 166 and 169).

Round 2

Reviewer 1 Report

Comments and Suggestions for Authors

All the comments have been addressed, and it can be accepted.

Reviewer 2 Report

Comments and Suggestions for Authors

Accept in present form.